# Genome-Wide Identification and Expression Analysis of Growth-Regulating Factors in *Eucommia ulmoides* Oliver (Du-Zhong)

**DOI:** 10.3390/plants13091185

**Published:** 2024-04-24

**Authors:** Ruoruo Wang, Ying Zhu, Degang Zhao

**Affiliations:** 1Plant Conservation Technology Center, Guizhou Key Laboratory of Agricultural Biotechnology, Biotechnology Institute of Guizhou Province, Guizhou Academy of Agricultural Sciences, Guiyang 550006, China; 2Ministry of Agriculture and Rural Affairs, Key Laboratory of Crop Genetic Resources and Germplasm Innovation in Karst Region, Guiyang 550006, China; 3The Key Laboratory of Plant Resource Conservation and Germplasm Innovation in Mountainous Region, Ministry of Education, Guizhou University, Guiyang 550025, China

**Keywords:** growth-regulating factor, miR396, *Eucommia ulmoides*, GA_3_, expression pattern

## Abstract

The roots, stems, leaves, and seeds of *Eucommia ulmoides* contain a large amount of trans-polyisoprene (also known as Eu-rubber), which is considered to be an important laticiferous plant with valuable industrial applications. Eu-rubber used in industry is mainly extracted from leaves. Therefore, it is of great significance to identify genes related to regulating the leaf size of *E. ulmoides*. Plant growth-regulating factors (GRFs) play important roles in regulating leaf size, and their functions are highly conserved across different plant species. However, there have been very limited reports on EuGRFs until now. In this study, eight canonical EuGRFs with both QLQ and WRC domains and two putative eul-miR396s were identified in the chromosome-level genome of *E. ulmoides*. It is found that, unlike AtGRFs, all EuGRFs contain the miR396s binding site in the terminal of WRC domains. These *EuGRFs* were distributed on six chromosomes in the genome of *E. ulmoides*. Collinearity analysis of the *E. ulmoides* genome revealed that *EuGRF1* and *EuGRF3* exhibit collinear relationships with *EuGRF2*, suggesting that those three genes may have emerged via gene replication events. The collinear relationship between *EuGRFs*, *AtGRFs*, and *OsGRFs* showed that *EuGRF5* and *EuGRF8* had no collinear members in *Arabidopsis* and rice. Almost all *EuGRFs* show a higher expression level in growing and developing tissues, and most *EuGRF* promoters process phytohormone-response and stress-induced cis-elements. Moreover, we found the expression of *EuGRFs* was significantly induced by gibberellins (GA_3_) in three hours, and the height of *E. ulmoides* seedlings was significantly increased one week after GA_3_ treatment. The findings in this study provide potential candidate genes for further research and lay the foundation for further exploring the molecular mechanism underlying *E. ulmoides* development in response to GA_3_.

## 1. Introduction

Growth-regulating factor (GRF) is a kind of plant-specific transcription factor that contains two highly conserved functional domains, namely QLQ and WRC, in the N-terminal regions of its protein sequences [1,2]. The QLQ domain in GRF protein is similar to a protein interaction domain located in the N-terminal region of the yeast SWI2/SNF2 protein and has been demonstrated to play a crucial role in its interaction with GIF protein [3,4]. The WRC domain is considered to function in the DNA-binding process due to the containing of a putative nuclear localization signal and a C3H-type zinc (Zn) finger motif [3,5]. Moreover, the WRC domains in most plant GRFs contain a conserved “RSRK-VE” motif at its terminus, which corresponds to a nucleotide sequence that complements the mature miR396 sequences [6,7]. Numerous studies have unequivocally demonstrated that miR396 can bind to its target sites on GRF to negatively regulate the expression of the *GRF* gene [8,9,10,11]. In addition to the QLQ and WRC domains, several other smaller conserved motifs were also found in the GRF protein sequences, i.e., FFD, TQL, and GGPL [5,12]. The first plant GRF, *OsGRF1*, was previously identified in deepwater rice with high expression levels in internode intercalary meristem in response to GA_3_ [3]. Currently, members of GRFs have also been identified in several other plant species: 9 in *A. thaliana*, 12 in *O. sativa*, 17 in *Brassica rapa*, 15 in *Lactuca sativa*, and 19 in *Populus alba × P. glandulosa* [5,6,7,13,14,15,16].

The development of leaves in the eudicot plant undergoes four stages. First, the founder cells are recruited in the peripheral zone of the shoot apical meristem to initiate leaf primordium. Secondly, distal growth occurs, and the polarities of the leaf (known as adaxial–abaxial and proximal–distal axes) are gradually formed. Third, the leaf blade separates from the petiole. Finally, the leaf area expands in multiple directions by cell proliferation and expansion [17]. Numerous transcription factors have been identified in promoting cell proliferation of developing leaves, including GRF [2,17]. The expression analysis of *GRFs* in various plant species has shown that *GRFs* were preferentially expressed in growing and developing tissues characterized by robust cell division, such as early flower buds, shoot tips with shoot apical meristem (SAM), and young leaves [5,12,13,14,18,19]. The overexpression of *AtGRF1*, *AtGRF2,* or *AtGRF5* in *A. thaliana* resulted in enhanced leaf growth and the development of transgenic plants with significantly larger leaves compared to those observed in the wild-type. Conversely, *Arabidopsis* plants with the loss function of those *AtGRFs* and overexpression of miR396s displayed reduced leaf size [4,5,20,21,22]. In maize, the high accumulation of *ZmGRF1* transcripts resulting from a mutation in the miR396s target sites led to an increase in the leaf blade size [23]. Additionally, the involvement of plant GRFs has also been demonstrated in various biological processes, including xylem development, reproductive organ development, flowering, stress response, and even plant longevity [2,24,25,26,27,28,29,30,31].

*E. ulmoides* is a kind of tertiary relict tree with great economic value [32]. The roots, stems, leaves, and fruits of *E. ulmoides* contain a significant quantity of Eu-rubber, which possess substantial industrial application value [32,33,34,35,36]. The current Eu-rubber used in industry is primarily extracted from the leaves of *E. ulmoides*. As reported, the majority of plant GRFs exhibit conserved functionality in positively regulating leaf blade expansion [2,15,37]. However, there have been very limited reports on the GRF transcription factors of *E. ulmoides* until now.

In this study, we identified eight canonical EuGRF transcription factors and two putative *eul-miR396s* in the chromosome-level genome of *E. ulmoides*. The collinearity between *EuGRFs* and *GRFs* of model plants *A. thaliana* and *O. sativa* was concurrently analyzed. Moreover, the expression of *EuGRF* genes and their corresponding promoter sequences were investigated. Via the above analysis, we found that almost all *EuGRFs* showed a higher expression level in growing and developing tissues, and the expression of most *EuGRFs* in junior leaves can be significantly induced by GA_3_ spraying within a three-hour timeframe. The findings presented in this study offer potential candidate genes for further investigation and lay the foundation for delving deeper into the molecular mechanisms underlying *E. ulmoides* development in response to GA_3_.

## 2. Materials and Methods

### 2.1. Plant Materials and Phytohormone Treatment

Four ages of *E. ulmoides* plants were used in this study. The 20-day-old seedlings were germinated on MS medium. The 2-month-old seedlings and 2-year-old plants were cultivated in appropriate pots filled with a peat soil mixture supplemented with perlite and vermiculite within a controlled climate chamber (16 h light/8 h dark, 22 ± 2 °C, 60% humidity). The 20-year-old plants were grown in a forestry station field at Guizhou University. For detecting the gene expression pattern of *EuGRFs*, the roots, hypocotyl, and cotyledon were collected from 20-day seedlings. The stems, shoot apical meristems with unopened leaves, and young leaves were collected from a 2-year-old plant. The female hybernacles, male hybernacles, female flower buds, and male flowers were collected from a 20-year-old female plant and a 20-year-old male plant, respectively. After sample acquisition, all tissues were immediately frozen in liquid nitrogen and stored in a −80 °C refrigerator. For gibberellin (GA_3_) treatment, 2-month healthy seedlings with 7–8 internodes were sprayed uniformly with 100 µM GA_3_ from a distance of approximately 10 cm above the plant, and the same volume of distilled water was used as control. The expression levels of *EuGRFs* induced by GA_3_ were analyzed at five time points (0 h, 0.5 h, 1 h, 2 h, 3 h). The initial juvenile leaves were selected for detecting the expression levels of *EuGRFs*. To measure plant height and internode length, a total of 9 plants were individually evaluated in both the experimental and control groups.

### 2.2. The Source of Original Data and Sequence Retrieval

The chromosome-level genome sequences of *E. ulmoides* were downloaded from the National Genomics Data Center (https://ngdc.cncb.ac.cn/?lang=zh accessed on 20 April 2024) with the access number GWHBISF00000000. The genome sequences of *A. thaliana* Araport11 and *O. sativa* Version 7.0 were acquired from the Phytozome 13 database (https://phytozome-next.jgi.doe.gov/ accessed on 20 April 2024). The gene IDs of *AtGRFs* and *OsGRFs* were obtained from the *Arabidopsis* gene family information on the Tair website (https://www.arabidopsis.org/browse/genefamily/index.jsp accessed on 20 April 2024) and the research article published by Choi et al. [13], respectively. Other plant GRF protein sequences were downloaded from PLAZA_v5_dicots (https://bioinformatics.psb.ugent.be/plaza/versions/plaza_v5_dicots/ accessed on 20 April 2024) with the gene family access number HOM05D000375 [38]. The sequences extract and ID rename were conducted using TBtools [39].

### 2.3. The Identification of EuGRFs

The Local BLAST and the Hidden Markov Model (HMM) search programs were used to identify the EuGRF members. The protein sequences of AtGRFs were utilized as queries to perform Local BLAST against *E. ulmoides* genome-wide protein sequences. The e-value of blastp was set as 1 × 10^−5^, and the resulting data were formatted in a tabular manner. After removing duplicate GRFs, the remaining GRFs were considered candidate EuGRFs, and their protein sequences were extracted using the Sequence Toolkit of TBtools. The candidate EuGRFs were analyzed using SMART (http://smart.embl-heidelberg.de/smart/set_mode.cgi?GENOMIC=1 accessed on 20 April 2024), and the potential conserved domains were identified via batch CD-Search Tool (https://www.ncbi.nlm.nih.gov/Structure/bwrpsb/bwrpsb.cgi accessed on 20 April 2024). Proteins containing both intact QLQ and WRC domains were designated as EuGRFs. The genome-wide HMM search was conducted using the Simple HMM Search tool of TBtools with default parameters. For the HMM search, the Pfam-A.hmm file was downloaded from InterPro (https://www.ebi.ac.uk/interpro/ accessed on 20 April 2024), and the Pfam IDs for conserved QLQ (PF08880) and WRC (PF08879) domains were obtained from Pfam (https://www.ebi.ac.uk/interpro/entry/pfam accessed on 20 April 2024). The common candidates identified via Local BLAST and HMM Search were recognized as true EuGRFs. These identified EuGRFs were named based on their sequence homology with AtGRFs.

### 2.4. The Identification of eul-miR396s

The miRNA sequences of all plant species were retrieved from the miRBase database (https://www.mirbase.org/ accessed on 20 April 2024). Subsequently, a Local BLAST search was conducted using the complete set of plant miRNA396s (Appendix A) as queries, allowing for a maximum of three mismatched bases. The resulting sequences were then subjected to extraction of 200 bp upstream and downstream fragments using the TBtools. Following the removal of repetitive sequences, RNA Folding Form (http://www.mfold.org/mfold/applications/rna-folding-form.php accessed on 20 April 2024) with the default parameters was employed to predict secondary structures of these sequences. Putative eul-miRNA396 precursors were identified based on their characteristic stem-ring structure, as indicated by the predicted results. Finally, RNAhybrid 2.2 software was utilized to analyze the interaction between eul-miRNA396 and *EuGRF* sequences, employing an energy threshold of −20.

### 2.5. Bioinformatic Analysis of EuGRFs

The sequence length, molecular weight, isoelectric point, and other basic physicochemical properties of EuGRF proteins were predicted using ExPASY (https://web.expasy.org/protparam/ accessed on 20 April 2024). The subcellular localization of EuGRFs was predicted using the web-server tools of Cell-PLoc 2.0 (http://www.csbio.sjtu.edu.cn/bioinf/Cell-PLoc-2/ accessed on 20 April 2024) and WoLF PSORT (https://wolfpsort.hgc.jp/ accessed on 20 April 2024).

### 2.6. Phylogenetic Analysis of GRFs

To construct the Neighbor-joining phylogenic tree, the amino acid sequences of AtGRFs, OsGRFs, and EuGRFs were aligned using MUSCLE v5 with the default parameters in MEGA6. The bootstrap replicates for the Neighbor-joining phylogenic tree were set to 1000, while the amino acid substitution model was the Poisson model. Gaps/missing data were set as complete deletion. The repeatability of constructing the Neighbor-joining phylogenetic tree was assessed via multiple iterations. To construct the Maximum Likelihood tree, the sequence alignment of GRF amino acid from 27 species was performed by MAFFT v7.487 with an auto-alignment strategy. After being trimmed by trimAl v1.2, the resultant alignment file was analyzed using iqtree v2.2.0. The best-fit model (JTT + F + I + G4) was chosen based on the Bayesian Information Criterion (BIC), and the bootstraps replicates as well as SH-like approximate likelihood ratio test (SH-aLRT) were set to 1000. The output tree file was visualized using FigTree v1.4.4 and further enhanced with Adobe Photoshop CC 25.6 for better presentation quality. The log file related to the construction of the Maximum Likelihood tree is enclosed within the Appendix A.

### 2.7. Chromosomal Location and Collinearity Analysis

The chromosomal location and collinearity analysis were performed using TBtools. Specifically, the Gene Location Visualize from the GTF/GFF v3.0 tool was utilized to display the chromosomal locations of *EuGRF* genes. The collinearity relationships were established using the One-Step MCScanx tool with parameters set at 5 hits for blastp and an e-value of 1 × 10^−10^. Advanced Circos tool was employed to visualize the collinearity within the *E. ulmoides* genome. To investigate the collinearity relationship among *E. ulmoides*, *A. thaliana*, and *O. sativa*, the Text Merge for MCScanx tool was used to integrate linked information and gene position data. Subsequently, the Multiple Synteny Plot tool was utilized to construct a triple-species collinearity relationship.

### 2.8. Total RNA Extraction and Real-Time Quantitative PCR

Total RNA from different samples was extracted using an RNA Easy Fast Plant Tissue Kit (Tiangen, DP452, Beijing, China) and dissolved in RNase-free water according to the manufacturer’s protocol. Subsequently, the first-strand cDNAs were synthesized by reverse transcription of 2 μg total RNA using the HiScript^®^ II 1st Strand cDNA Synthesis Kit (+gDNA wiper; R212; Vazyme, Nanjing, China). The quantified cDNA derived from a 200 ng RNA template was employed as a template for qRT–PCR analysis. qRT-PCR was conducted using an SYBR qPCR Mix (Tsingke, TSE401, Beijing, China) on a Real-Time PCR System (ABI QuantStudio3, Thermofisher, Waltham, MA, USA). Gene-specific primers were designed using the Primer-BLAST tool (https://www.ncbi.nlm.nih.gov/tools/primer-blast/ accessed on 20 April 2024), and their specificity was confirmed via sequence alignments and melting curve analysis via qRT–PCR. Appendix A provides information about the primer sequences. The qRT-PCR conditions included an initial polymerase activation step at 95 °C for 5 min followed by 40 cycles of denaturation at 95 °C for 10 s and annealing at 60 °C for 30 s. Additionally, a dissociation curve was generated to validate product specificity according to the PCR system manufacturer’s recommendations. Finally, relative transcript levels of amplification products were calculated using the 2^−∆Ct^ method, with *EuACTIN* serving as an internal reference gene.

### 2.9. Construction of the Phylogenetic Tree of Plant Species

The phylogenetic tree of plant species was analyzed using NCBI Taxonomy Tools, and the estimation of species divergence times was assessed using TimeTree5. The plant species utilized in this study are listed in Appendix A.

## 3. Results

### 3.1. The Characterization of EuGRFs in E. ulmoides

GRF is a pivotal transcription factor that plays crucial roles in plant growth, reproductive development, and response to external stimuli [1,2]. To identify GRF members in *E. ulmoides*, a genome-wide Local BLAST search was performed. By employing AtGRFs protein sequences as queries for blastp analysis, we obtained 12 candidate proteins (Appendix A). According to the analysis of the conserved domains in these 12 candidate proteins, we observed that only 8 proteins possess both conserved QLQ and WRC domains, namely GWHPBISF018630, GWHPBISF001116, GWHPBISF022031, GWHPBISF008846, GWHPBISF020960, GWHPBISF018381, GWHPBISF000663, and GWHPBISF002516 (Figure 1A,B and Appendix A). Additionally, a comprehensive genome-wide HMM search was performed to identify EuGRFs. By utilizing seed sequences for the WRC domain and QLQ domain as queries, we successfully identified 11 and 16 candidate proteins, respectively, of which 8 members were found to be common with the candidates identified by Local BLAST (Appendix A). Therefore, these 8 candidate proteins were identified as EuGRFs. Then, we conducted the subsequent analysis.

### 3.2. The Identification of miR396s in E. ulmoides and the Analysis of Their Target Binding Sites in the EuGRFs

The post-transcriptional regulation of most plant GRFs can be mediated by miR396 [2]. By analyzing the protein sequences of EuGRFs, we observed that all these proteins contain the conserved “RSRK-VE” motif at the terminal of the WRC domain, which corresponds to the target sites of miR396s (Figure 1B). To investigate whether *EuGRFs* are regulated by miR396s in *E. ulmoides*, we also identified potential eul-miR396s in the *E. ulmoides* genome. Using mature sequences of plants miR396s as queries for blastn analysis, we obtained 38 candidate eul-miR396s sequences with less than three mismatched bases (Appendix A). In plants, miRNA biosynthesis begins with transcription of the *MIR* gene that possesses canonical stem-loop structure within their intramolecular bases [40]. Therefore, we extracted the 38 candidate eul-miR396s sequences along with their upstream and downstream regions spanning 200 bp and analyzed their secondary structures. Our results revealed that only two sequences could form stable stem-loop secondary structures within their intramolecular bases (Appendix A). Based on the similarity to ath-miR396 sequences, we designated these two candidate eul-miR396 sequences as eul-miR396a and eul-miR396b. The corresponding precursors forming stable stem-loop secondary structures were named *Pre-miR396a* and *Pre-miR396b* (Figure 1C,D).

### 3.3. Phylogenetic Analysis and the Classification of EuGRFs

To investigate the phylogenic relationship of EuGRFs, we performed protein sequence alignment and phylogenetic analysis using full-length protein sequences of 8 EuGRFs, 9 AtGRFs, and 12 OsGRFs. Based on the Neighbor-Joining phylogenetic tree (Figure 2A), all GRFs were divided into six distinct groups (Group I to Group VI). Specifically, EuGRF1, EuGRF2, and EuGRF3 belonged to Group I; EuGRF4 belonged to Group III; EuGRF5 and EuGRF6 belonged to Group V; and EuGRF7 and EuGRF8 belonged to Group VI. Based on the homologous relationships of EuGRFs with AtGRFs, we named the EuGRFs EuGRF1, EuGRF2, EuGRF3, EuGRF4, EuGRF5, EuGRF6, EuGRF7, and EuGRF8, respectively (Table 1). Apart from the conserved QLQ and WRC domains in GRFs’ N-terminal region, several smaller conserved motifs were also observed in the C-terminal region of plant GRFs. In order to characterize the EuGRFs comprehensively, we analyzed the conserved motifs of all 8 EuGRFs along with 9 AtGRFs and 12 OsGRFs. Consistent with our phylogenetic analysis results (Figure 2B,C), it was found that both the TQL motif (Motif 3) and FFD motif (Motif 4) were present in GRFs belonging to Group III and Group VI. In the GRF members of *E. ulmoides*, EuGRF1 and EuGRF2 exclusively possess a TQL motif in their C terminals regions, whereas other members of EuGRFs possess both TQL and FFD motifs (Figure 2B,C).

Meanwhile, we conducted an analysis of the nucleotide sequences of eight *EuGRF* genes to ascertain the presence of miR396s target sites. The results revealed that all the corresponding coding sequences of *EuGRFs* exhibited nearly perfect complementarity with putative eul-miR396s, demonstrating a minimum free energy hybridization lower than −33 kcal/mol (Figure 2D, Appendix A). Specifically, *EuGRF1*, *EuGRF2*, *EuGRF3*, *EuGRF4*, *EuGRF5*, *EuGRF6* and *EuGRF7* displayed complete complementary bases at the 5′ and 3′ ends with 12 and 7 bases, respectively. In contrast, the miR396s binding site of *EuGRF8* had completely complementary bases at its 5′ and 3′ ends with 11 and 8 bases, respectively (Figure 2D, Appendix A).

### 3.4. Chromosome Location and Gene Duplication of EuGRFs in E. ulmoides Genome

To determine the distribution of *EuGRF* genes in the *E. ulmoides* genome, we visualized the genomic annotation to analyze their distribution and location on each chromosome. The results showed that eight *EuGRFs* are distributed across six chromosomes in *E. ulmoides* genome. Specifically, *EuGRF1*, *EuGRF2*, *EuGRF4*, and *EuGRF5* were located on chromosomes 15, 1, 11, and 10, respectively. *EuGRF3* and *EuGRF7* were located on chromosome 8, while *EuGRF6* and *EuGRF8* resided on chromosome 6 (Figure 3A). Genome-wide duplication events play a crucial role in the evolutionary dynamics of plants, as most terrestrial plant species have experienced at least one such event. Gene replication often accompanies these duplication events, leading to subsequent processes of gene subfunctionalization and neofunctionalization. Based on the genomic information of *E. ulmoides*, it has undergone two whole-genome duplication events during its evolutionary history [41]. To investigate whether members of the *EuGRF* genes have also undergone gene replication accompanying *E. ulmoides*’ genome duplication events, we conducted a colinear analysis of genes within its genome. The results revealed that out of a total of 25,477 genes analyzed in the *E.ulmoides* genome (Appendix A), approximately 14.53% (3703) showed collinear relationships indicating potential gene replication occurrences during its evolution process (Figure 3B). Notably, *EuGRF2* exhibited collinearity with both *EuGRF1* and *EuGRF3* (Figure 3B), suggesting that two genome-wide replication events in *E. ulmoides* genome might have facilitated the amplification within the GRF gene family, particularly for members belonging to Group I.

### 3.5. The Collinearity Analysis of GRFs in E. ulmoides, A. thaliana, and O. sativa

Collinearity analysis of gene family members across different species can provide insights into the distribution and arrangement of homologous genes, particularly those associated with plant growth and development. We conducted a collinearity analysis using the genome of *A. thaliana*, *E. ulmoides*, and *O. sativa* to determine the collinearity of *GRF* genes among these species and predict potential gene functions for *EuGRFs*. The analysis revealed eight linked pairs of *GRFs* between *A. thaliana* and *E. ulmoides* genomes (Appendix A and Figure 3C), as well as six linked pairs between *E. ulmoides* and *O. sativa* genomes. Notably, both *A. thaliana* and *O. sativa* genomes lacked collinear members with *EuGRF5* and *EuGRF8*. However, *EuGRF7* showed collinearity with two members (*OsGRF3* and *OsGRF4*) in the *O. sativa* genome, similar to the pattern observed for *EuGRF4* in the *A. thaliana* genome (Figure 3C). Taken together, these findings suggest a substantial gene collinear relationship in *GRFs* between *E. ulmoides* and *A. thaliana,* as well as *O. sativa*.

### 3.6. Expression Pattern Analysis of EuGRFs in Different Organs of E. ulmoides

Numerous studies have demonstrated a strong correlation between gene function and gene expression patterns in plants. To predict the possible functions of *EuGRFs* in the growth and development of *E. ulmoides*, we collected tissues from different developmental stages and performed qRT-PCR to analyze the expression levels of *EuGRF* genes. The results revealed distinct tissue-specific expression patterns for all *EuGRF* genes (Figure 4). Particularly, *EuGRF1* and *EuGRF2*, as well as *EuGRF7* and *EuGRF8,* exhibited similar expression profiles with significantly higher levels observed in SAM with young leaves of 20-day seedlings, winter buds of male and female plants aged 20 years, as well as female flower buds (Figure 4). Among these analyzed tissues, the highest expression levels were found for *EuGRF1*, *EuGRF3*, and *EuGRF5,* followed by *EuGRF*2, *EuGRF4*, *EuGRF6,* and *EuGRF7*. On the other hand, *EuGRF8* showed relatively lower expressions across all detected tissues. Furthermore, negligible or undetectable expressions were observed for *EuGRF4*, *EuGRF5*, and *EuGRF6* in roots, hypocotyls, cotyledons of 20-day seedlings, and stems of 2-year plants. Additionally, *EuGRF4* expression was barely detected in male flowers (Figure 4). Previous results indicated that there are collinear relationships among members within the subfamily containing *EuGRF1*, *EuGRF2*, and *EuGRF3* (Figure 3B). Notably, gene expression profile analysis revealed that *EuGRF1* and *EuGRF3* exhibited significantly higher expression levels than *EuGRF*2 across all analyzed tissues (Figure 4), suggesting a potentially more prominent functional role played by *EuGRF1* and *EuGRF3* in the growth and development processes of *E. ulmoides* than *EuGRF2*.

### 3.7. Cis-Element Analysis of EuGRFs

The cis-elements located on gene promoters play crucial roles in regulating gene expression. To explore potential cis-elements within the promoter region of *EuGRF* genes, a 3.5 kb sequence upstream of the start codon was extracted and analyzed for the presence and distribution of putative cis-elements. According to the result of putative cis-elements analysis, it was found that distinct cis-elements exhibited specificity in their distribution and abundance among *EuGRF* promoters (Appendix A, Figure 5A). Based on the distribution of cis-elements, *EuGRFs* can be clustered into two clades, clade I (*EuGRF6*, *EuGRF5*, *EuGRF3*, *EuGRF7*) and clade II (*EuGRF1*, *EuGRF2*, *EuGRF4*, *EuGRF8*) (Figure 5A). Analysis of the distribution of cis-elements showed a predominant enrichment of cis-elements related to meristem expression within the promoter region of *EuGRF6*. Elements associated with auxin response, GA response, ABA response, and salicylic acid response were predominantly distributed on the promoters of *EuGRF1*, *EuGRF5*, *EuGRF4*, and *EuGRF5*, respectively (Appendix A, Figure 5B). Besides, stress-responsive cis-elements were also identified in promoters of *EuGRFs* with varying degrees depending on each specific element type. Specifically, *EuGRF8*, *EuGRF1*, *EuGRF8,* and *EuGRF4* exhibited a higher abundance of cis-elements associated with “defense and stress responsiveness”, “anoxic-inducibility”, “low-temperature responsiveness”, and “drought-inducibility”, respectively (Figure 5A,B). Additionally, cis-elements associated with “light-responsive”, “circadian control”, “zein metabolism regulation”, “seed-specific regulation”, and “flavonoid biosynthetic genes regulation” were also found to be present in *EuGRF* promoter regions (Appendix A, Figure 5A).

### 3.8. The Expression of EuGRFs in E. ulmoides Seedlings in Response to GA3 Treatment

The cis-element analysis results revealed that the promoter regions of most *EuGRFs* contain varying quantities of cis-elements associated with GA_3_ response. To verify whether the expression of *EuGRFs* can be induced by GA_3_ treatment, we examined the expression levels of *EuGRFs* in the initial juvenile leaves of *E. ulmoides* seedlings after 0 h, 0.5 h, 1 h, 2 h, and 3 h of GA_3_ treatment. The results showed that, except for *EuGRF8*, the expression levels of other *EuGRFs* were significantly induced following a 0.5 h induction (Figure 6). Furthermore, we observed that the expression of *EuGRFs* can be induced after a 0.5 h treatment with GA_3_ (Figure 6). Subsequently, we continuously monitored and measured plant height at four time points: 1 day, 3 days, 5 days, 7 days, and 11 days after spraying GA3 onto seedlings. The results indicated that GA_3_ application significantly stimulated the elongation of plant height of *E. ulmoides* seedlings (Figure 7A,B). By measuring internode lengths in seedlings treated with GA3 or control solution (water), we found that GA_3_ treatment significantly enhanced growth in nascent internodes (1st–3rd), as well as developing internodes (−1st and −2nd) in *E. ulmoides* seedlings under GA_3_ treatment conditions (Figure 7C–G). There was no significant increase in the length of the −3rd, −4th, and −5th internodes (Figure 7H–J). These findings provide evidence that GA_3_ treatment can effectively induce the expression of *EuGRF1*, *EuGRF2*, *EuGRF3*, *EuGRF4*, *EuGRF5*, *EuGRF6*, and *EuGRF7* in young leaves of *E. ulmoides* seedlings, and which might contribute to promoting growth specifically within nascent and developing internodes.

## 4. Discussion

Numerous studies have demonstrated that the first GRF transcription factor emerged in streptophytas [42,43,44,45,46]. Only a limited number of GRFs have been identified in the genomes of basal lineages of land plants, commonly known as bryophytes or moss (one in the *Marchantia polymorpha* genome and two in the *Physcomitrella patens* genome) [43]. Additionally, four GRFs were identified in the genome of the pteridophyte plant *Selaginella moellendorffii* [43,46]. Previous studies have reported that GRF transcription factors underwent expansion via whole-genome duplication events, chromosome segmental replication, and gene duplication during plant evolution [1,12,16]. In the phylogenetic tree established previously, plant GRF transcription factors were classified into six groups [1]. In this study, we identified eight EuGRF transcription factors in the *E. ulmoides* genome, which also clustered into six groups. Notably, Group I and Group V were the common group in the members of GRFs in *A. thaliana*, *O. sativa*, and *E. ulmoides*. To investigate whether members from groups I and V are conserved across all terrestrial plants’ genomes, we utilized GRF family members of 26 terrestrial plants downloaded from the PLAZA database (https://bioinformatics.psb.ugent.be/plaza/versions/plaza_v5_dicots/ accessed on 20 April 2024) to construct a maximum likelihood phylogenetic tree (Appendix A) and then counted the distribution pattern of group members across different species (Figure 8). The results showed that at least one member belonging to group I is present in all analyzed terrestrial plants, while members belonging to group V are specifically found within vascular plant (gymnosperms and angiosperms) genomes (Figure 8). Therefore, it can be speculated that GRFs from group Ⅰ may be the earliest emerged GRFs in terrestrial plants, and members from group V might play a crucial role in plant development and terrestrial adaptation for the vascular plant.

In *A. thaliana*, despite the presence of some overlapping functions among members of AtGRFs, other members of AtGRFs were unable to replace the role of AtGRF5 (a member of group V) [1,4]. In contrast to *AtGRF1* and *AtGRF2*, the single gene mutant of *AtGRF5*, *atgrf5*, develops a smaller leaf compared to the wild type due to a decrease in cell number [4]. Consistent with previous reports, we observed that *GRFs* belonging to group V (homologs of *AtGRF5*) display the highest expression levels across different species [12,14]. Moreover, homologs of *AtGRF5* are conserved in various eudicots species, irrespective of their herbaceous or woody nature, particularly in terms of leaf development [4,16,28]. In *E. ulmoides*, *EuGRF5*, *a* homolog of *AtGRF5,* also exhibits the highest expression levels in the juvenile tissues compared to other *EuGRFs*, indicating that *EuGRF5* might play a central role in leaf development in *E. ulmoides*.

The coding sequences of plant *GRFs* universally contain miR396 binding sites, which correspond to the “RSRK-VE” motif within their protein WRC domains [1,2,47]. However, it should be noted that not all plant GRFs possess binding sites for miR396s. For instance, in *A. thaliana*, *AtGRF5* and *AtGRF6* do not exhibit complementarity to the mature miR396s sequences [1,2,21]. Similarly, homologs of *AtGRF5/6* in *Brassica campestris* and *Brassica rapa* also lack the conserved “RSRK-VE” motif in their coding sequences and do not have the target binding sites for miR396s [7,14]. Interestingly, *OsGRF11* and its homologs *ZmGRF4/10* belong to distinct subgroup members of *AtGRF5/6* and do not possess the conserved “RSRK-VE” motif either [13,48]. In this study, all members (including *EuGRF* genes in our unpublished genome, Appendix A) of identified *EuGRFs* exhibited the conserved “RSRK-VE” motif, and their corresponding coding sequences demonstrated near-perfect complementarity with putative eul-miR396s by displaying a minimum free energy hybridization lower than −33 kcal/mol (Appendix A).

The *GRF* gene in plants was initially identified in rice, and the previous study has shown that the expression of *OsGRF1* in intercalary meristem can be induced by GA_3_ [3,13]. However, it has been observed that the expression levels of *AtGRFs* remained unresponsive to GA_3_ induction [5]. In this study, we have demonstrated a significant induction of *EuGRF1*, *EuGRF2*, *EuGRF3*, *EuGRF4*, *EuGRF5*, *EuGRF6,* and *EuGRF7* expression in juvenile leaves after the application of GA_3_. The expression level of *EuGRF8* was unaffected by GA_3_ treatment. Nevertheless, promoter regulatory element analysis revealed the presence of a GA_3_ response element on the *EuGRF8* gene promoter. Apart from these response elements on the promoter region, gene expression regulation is also influenced by additional factors such as silencers and other regulatory elements located proximal to the gene. Therefore, although GA_3_ response elements are present on the promoter of *EuGRF8*, their expression level remains unresponsive to GA_3_ stimulation, potentially attributed to the presence of other inhibitory regulatory elements. Furthermore, no GA_3_ response element was found on the promoter region of *EuGRF2*; however, the expression of *EuGRF2* was induced by GA_3_, potentially indicating an indirect regulatory mechanism governing *EuGRF2* expression in response to GA_3_. It is also possible that the location of the GA_3_ response element on *EuGRF2* lies beyond the confines of its 3.5 kb promoter region (e.g., introns, 3′ UTR, etc.).

## 5. Conclusions

In this study, we successfully identified eight *EuGRFs* and two putative eul-miR396 in the chromosome-level genome of *E. ulmoides*. The complementation analysis results demonstrated that eul-miR396s exhibit nearly perfect complementarity with all members of *EuGRF* genes, indicating potential interactions between eul-miR396s and *EuGRFs*. Furthermore, we conducted an expression analysis of *EuGRFs* and investigated their induction by GA_3_. These findings provide a repertoire of potential candidate genes for further investigation and lay the foundation for elucidating the molecular mechanisms underlying *E. ulmoides* development in response to GA_3_.

## Figures and Tables

**Figure 1 plants-13-01185-f001:**
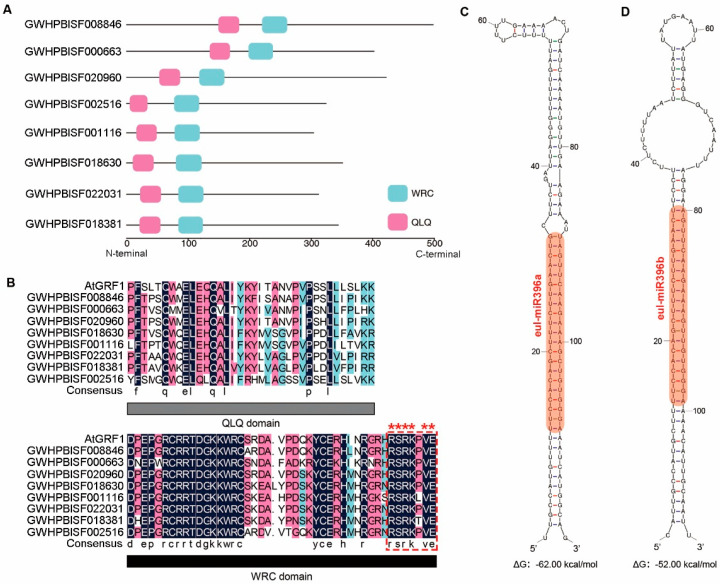
Protein sequence characteristics of EuGRF members. (**A**) Physical locations of conserved domains in the 8 EuGRF protein sequences annotated in the *E. ulmoides* genome. (**B**) The sequence alignment of QLQ and WRC domains in EuGRFs. The amino acids labeled by asterisks within the red dotted box represent the conserved “RSRK-VE” motif, whose corresponding nucleotide sequences serve as miR396 targets. (**C**,**D**) Hairpin structures formed by pre-miR396s. The orange areas denote the location of the miR396s.

**Figure 2 plants-13-01185-f002:**
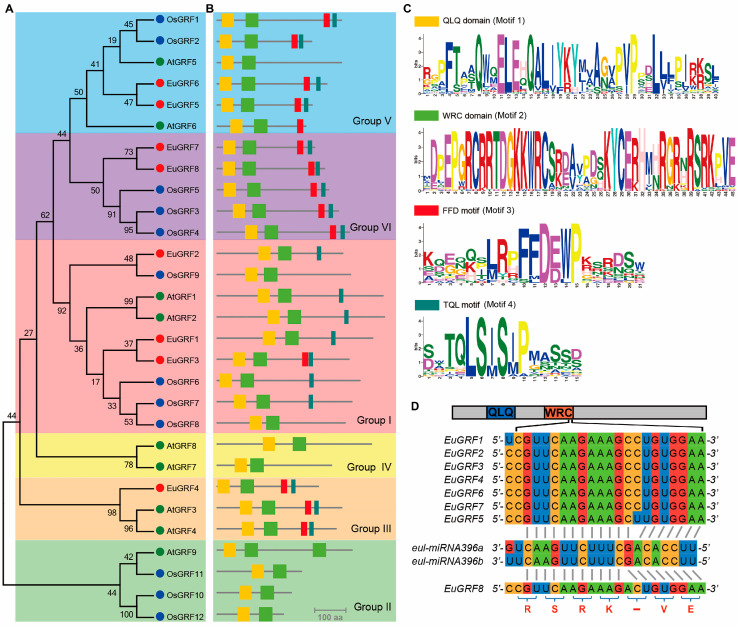
Phylogenetic relationship among the complete protein sequences of *E. ulmoides*, *A. thaliana*, and *O. sativa* GRF proteins. (**A**) The Neighbor-joining phylogenic tree of AtGRFs, OsGRFs, and EuGRFs constructed by MEGA6.0. The physical locations of conserved motifs in GRF proteins and their Seqlogo of position weight matrix are depicted in (**B**,**C**), respectively. Motif 1, 2, 3, and 4 correspond to QLQ domain, WRC domain, FFD motif, and TQL motif. (**D**) Diagram of the eul-miR396s target sites of EuGRFs. The conserved QLQ and WRC domains in GRF protein are highlighted in blue and orange, respectively. The amino acids “RSRK-VE” correspond to the target sites for eul-miR396s.

**Figure 3 plants-13-01185-f003:**
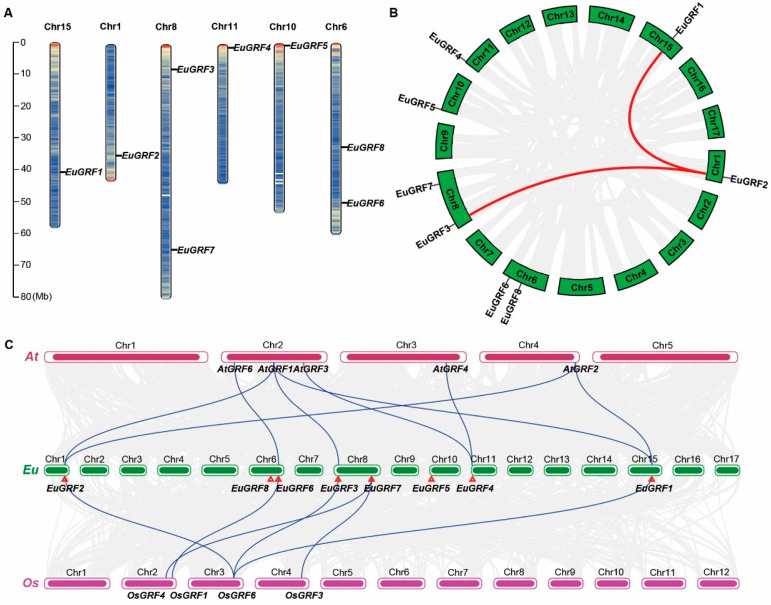
The physical location of *EuGRFs* on chromosomes and collinearity analysis of *EuGRFs*. (**A**) The physical locations of *EuGRFs* on the chromosomes of the *E. ulmoides* genome. The scale bar on the left represents the chromosome length. Mb, mega-base. (**B**) Collinear relationships of annotation genes within the *E. ulmoides* genome. The highlighted lines represent the collinearity of *EuGRF2* with *EuGRF1* and *EuGRFs*, respectively. The gray lines show the collinear blocks of annotation genes in the *E. ulmoides* genome. (**C**) The collinearity relationship between annotation genes of *E. ulmoides*, *A. thaliana*, and *O. sativa* genomes. The gray lines display the collinear blocks between *E. ulmoides* and other species, and the highlighted lines indicate the collinearity of *EuGRFs* with *AtGRFs* and *OsGRFs*. The red triangles denote the physical locations of *EuGRFs*. Chr, Chromosome.

**Figure 4 plants-13-01185-f004:**
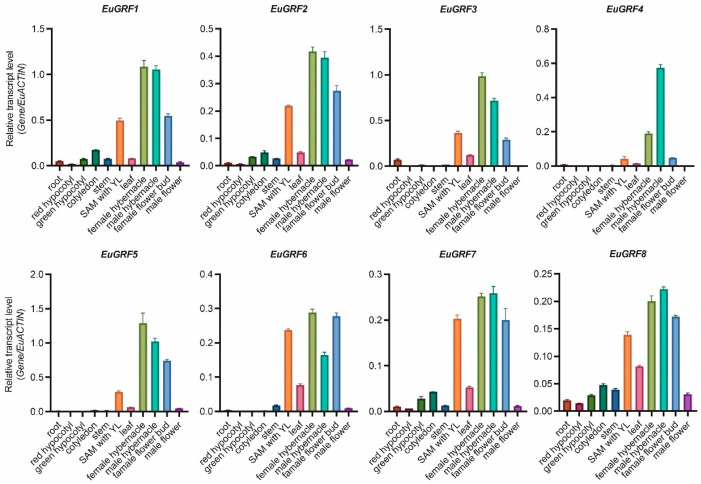
Expression patterns of *EuGRFs* in different tissues of *E. ulmoides*. The expression levels of all *EuGRFs* were determined by qRT-PCR. The *EuACTIN* gene was used as the internal control. The data are shown as the mean values ± SD (n = 3), and similar expression patterns were obtained using an alternative internal control gene, *EuEF1α*.

**Figure 5 plants-13-01185-f005:**
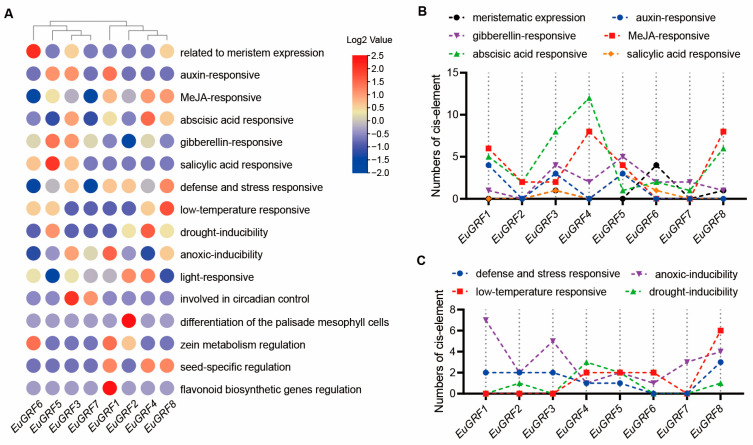
Cis-element analysis within the 3.5 kb promoter sequences of *EuGRFs*. (**A**) Clustering heatmap of cis-elements in the promoter regions of *EuGRFs*. (**B**) The counting of meristematic expression and hormone-responsive cis-regulatory elements in the promoters of *EuGRFs*. (**C**) The counting of stress-response cis-regulatory elements in the promoters of *EuGRFs*.

**Figure 6 plants-13-01185-f006:**
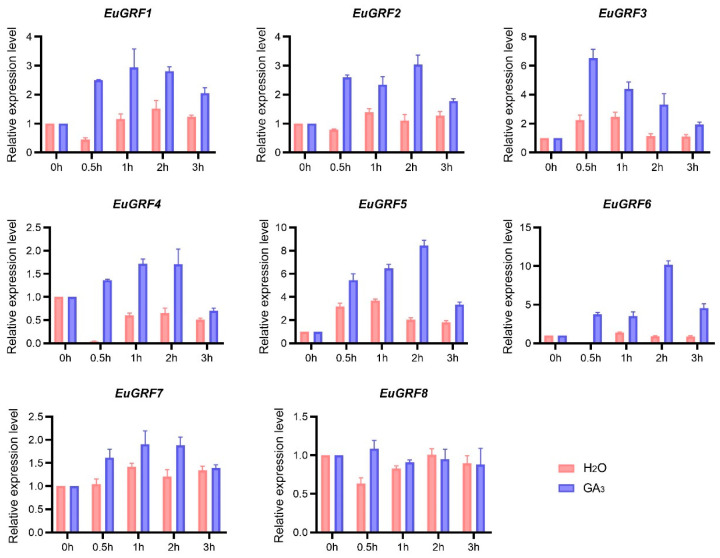
Reactive gene expression of *EuGRFs* in the initial juvenile leaves upon treatment with GA_3_. The control group was established by treating it with distilled water. The gene expression levels were quantified using qRT-PCR, with *EuACTIN* serving as the internal control. Two independent experiments showed similar results.

**Figure 7 plants-13-01185-f007:**
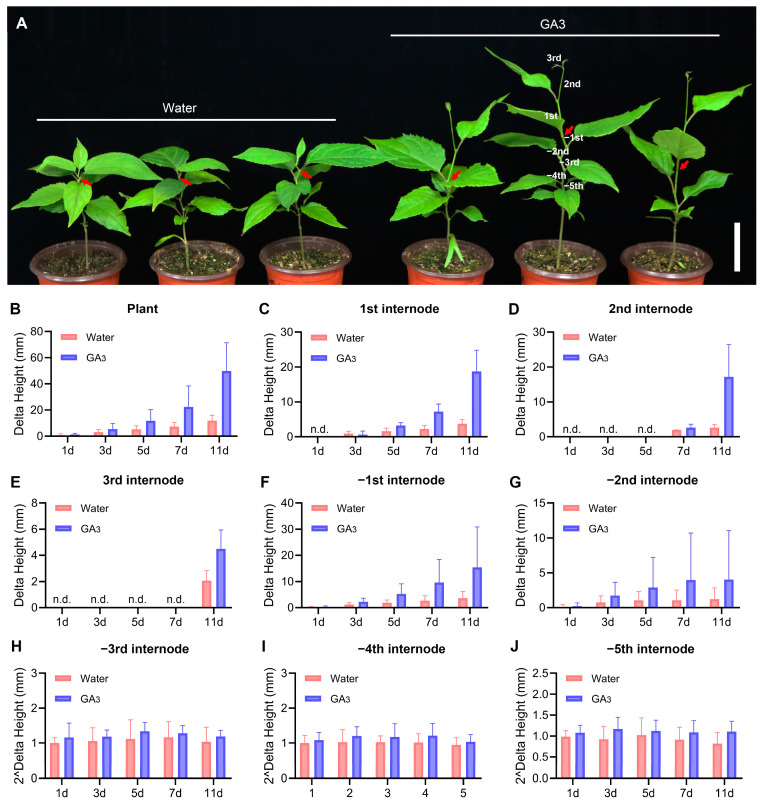
Phenotypic analysis of E. ulmoides following GA_3_ treatment. (**A**) The phenotype of *E. ulmoides* seedlings following a 15-day treatment with GA_3_. Each group consists of three independent 2-month-old healthy seedlings. The length of the scale bar in the bottom right corner is 5 cm. (**B**) The plant height of seedlings after GA_3_ treatment for 1, 3, 5, 7, and 11 days. (**C**–**J**) The delta height (the elongation height relative to the pre-treatment height) of different internodes following GA_3_ treatment. The positions of 1st, 2nd, 3rd, −1st, −2nd, −3rd, −4th and −5th internodes are presented in (**A**).

**Figure 8 plants-13-01185-f008:**
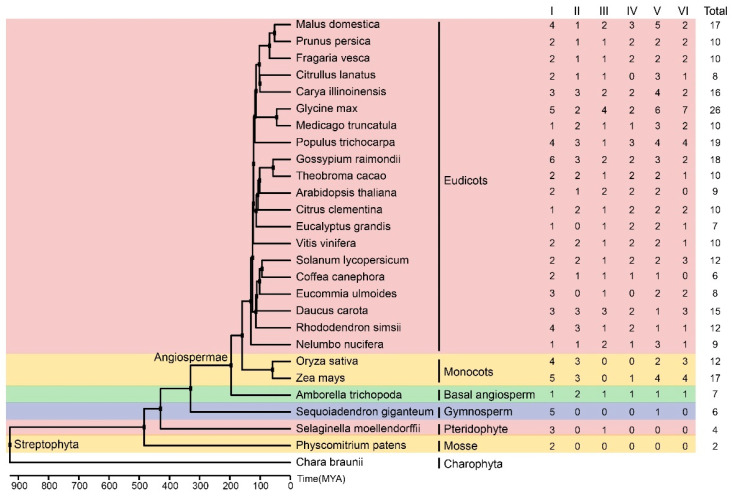
The phylogenetic tree of 27 plant species and the distribution patterns of GRFs in different subgroups. The species divergence times are shown at the base of the phylogenetic tree. MYA, million years ago. The numbers on the shade represent the count of GRFs in each subgroup of plants. The numbers on the right represent the cumulative GRFs in different plant species.

**Table 1 plants-13-01185-t001:** The basic physicochemical properties and predicted subcellular localizations of putative EuGRFs.

	Protein ID	Arabidopsis Homolog	Protein Length	Molecular Weight	Theoretical pI	Subcellular Localization
EuGRF1	GWHPBISF008846	AtGRF1, 2	498	53,478.65	8.14	Nucleus
EuGRF2	GWHPBISF000663	AtGRF1, 2	402	44,614.49	9.31	Nucleus
EuGRF3	GWHPBISF020960	AtGRF1, 2	422	45,306.59	8.27	Nucleus
EuGRF4	GWHPBISF002516	AtGRF3, 4	324	34,645.58	8.78	Nucleus
EuGRF5	GWHPBISF001116	AtGRF5, 6	304	33,933.86	8.24	Nucleus
EuGRF6	GWHPBISF018630	AtGRF5, 6	351	39,978.65	8.42	Nucleus
EuGRF7	GWHPBISF022031	—	312	34,787.05	8.65	Nucleus
EuGRF8	GWHPBISF018381	—	344	38,022.43	8.34	Nucleus

## Data Availability

Data are contained within the article and Appendix A.

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
