# Peer review of "Genome-Wide Identification and Expression Analysis of Growth-Regulating Factors in Eucommia ulmoides Oliver (Du-Zhong)"

_plants, 2024, doi:10.3390/plants13091185_

Round 1
Reviewer 1 Report
Comments and Suggestions for Authors
Comments and Suggestions for Authors
Dear Author,
It is my pleasure to review the manuscript entitled “Genome-wide identification and expression pattern analysis of grouth-regulating factors in Eucommia ulmoides” a research article submitted to MDPI Journal, Plants. Authors of this manuscript identified and characterized eight EuGRF transcription factors in the E. ulmoides. Authors have characterized physical and chemical properties, gene structure, phylogeny, and expression patterns after GA3 treatment through a series of bioinformatic and lab experiments. Overall, the experiments, they performed, are well and the results are convincing. Thus, the presented results take up an important topic consistent with the profile of the Journal.
However, I have some suggestions, which might improve the manuscript to make important to the wider readers.
· Improvement in English is very necessary for clear understanding
· I have corrected some point in the main text, please follow them as well
· Article similarity index should be less than 20%
Title: you may use local name aside
Abstract: Good organization with results order, but not matching with journal style
-Elaborate GRF at their first time use
-Gene name should be italic throughout the text.
Introduction:
-Reference style is not as the journals’ style
-Many other TF also play important role in growth and development. You may discuss a few
-Introduction should be more constructive with rationale of the study. Elaborate clearly, why this research is necessary. Here huge bioinformatic experiments have been done, those should have some reflection in introduction. Specific and research results oriented introductory is needed rather than vey general discussion
Materials
- Here, 20D, 2Y and 10Y samples are used. Are those from chamber? MS media? Or field? Not clear. Material preparation should be clear and informative enough. For further research, do we need to wait up to 10 years? For GA3 treatment, where the plants were grown?
- Please mention, target genes’ type and name, and how the primers were designed.
- Need to detail description of phylogenetic tree construction procedure for reproducibility.
Results
Mention the stage of ample collection.
-Conclusion mostly based on methodology. Improve substantially using obtained results conclusion
My Recommendation about this article

Comments on the Quality of English LanguageMinor editing of English language required
Author Response
Dear Reviewer,
We really appreciate you for reviewing the manuscript and giving back your valuable comments which make this manuscript to be more perfect. All of these comments have contributed a lot to improve the quality of this manuscript.
For clarity, all comments and suggestions have been replied one by one and addressed in the revision. The detailed point-by-point responses are listed in the attached file. For clarity, all comments are given with red letters and responses are given with the blue letters below. If there are any other modifications we could make, please feel free to let us know.
Best regards,
Ruoruo Wang

Reviewer 2 Report
Comments and Suggestions for Authors
Studying transcription factors families in new species is always interesting and in this case the plant have also economical interest. The difficult of working with non model species algo increments difficulty and must taken into account. In this sence, I think the research authors presents is interesting. However, I have found some important issues about interpretation of results and too much speculation with just predicted data. Also, I think there are some experiments that can be conducted to better support the results and discussion that I guess it must not be difficult to do.
I present the minor and major issues:
Page 2 Line 10 : ginnerellins > gibberellins
Page 2 Line 11 : suggestion : “Currently, variable and variable number of GRF genes have been identified in different plant species: 9 in A. thaliana, … “
Page 2 Line 20 : in regulate > in regulating
Page 2 Line 33-34 : In Li et al 2022 is not perfectly demonstrated the properties authors indicate, actually is a bibliography review indicating some mechanism can be associated with some of the components found in the plant. Although it can have those properties, al estudies only suggest some correlations and at that moment is still too early for absolute affirmations, we are scientist and tll the moment is perfectly demonstrated with enough number of individual, all controls and double controls, and discarded placebo effect, we can only suggest or show that there are indications but never absolute affirmations.
M&M 2.1 : Line 12 > “to detect” or “for detecting” but not “to detected”
M&M 2.1 : I don’t see what is the interest of a climate chamber controlled by a IA… just because is now on trend? Just for controlling photoperiod and temperature you don’t need an IA. By contrast it is requerid to give the chamber data (Brand, model, etc., both for the chamber and for lights (if they are independently adquired) so experiments can be repeated or have the correct information of kind of light. Actually is not indicated the intensity of light, neither. And if an IA is been used, data about with neuronal net is used, etc., must be indicated.
3.1 Last line: anlalysis > analysis [There are enough mistakes to suggest a carefull revision of text to correct them… you can even use IA]. I will not indicate any more, but they exist.
For some reason I have been unable to open the supplementary zip file, so maybe I do some questions that are well represented in that data.
3.4 Suggestion about order duplications of members 1, 2 and 3 is wrong. First at all, authors must remember that when a duplication event occurs both copies are identical and it is difficult to stablish which genome location was the original one. From phylogenetic tree it is seen that 1 and 3 are the more closely related, meaning they came from the more recent duplication event (actually in Figure 3B conection between 1 and 3 must be marked in red as well, and it must be even strongest). At that point it means that only one copy, we don’t know if in chr15 or in chr8, exist before last duplication. In that moment, the two more related copies was this ancestral member “pre-1-3” and member 2. This two genes came from one unique ancestral copy but we can’t know if it was in chr1 or in chr8 or in chr15, as when this ancestral copy duplicated, both copies were identical. Without more information from very close related species where to find other number of copies (for instance a related species with only 2 copies that can be located by collinearity with two of the chromosomes) it is imposible to know the evolutive path. Case 1: GRF ancestral copy is in Chr15 and duplicates to chr1, after some time they accumulate changes and then the copy in chr15 duplicates again to chr8. Case2: GRF ancestral copy is in Chr1 and duplicates to chr8, after some time they accumulate changes and then the copy in chr8 duplicates to chr15. Case 3: GRF ancestral copy is in Chr8 and duplicates to chr1, after some time they accumulate changes and then the copy in chr8 duplicates again to chr15. In all case the more closely related copies (more recent duplication) is between chr8 and chr15. In fact, what is absolutely imposible is that the origin of members 1 and 3 was member 2, because in this case the more related two copies must contains member 2 and one of the others, the one from the last duplication.
3.4: With so different collinearities between species and intraspecies, is just speculative any functional interpretation of different copies, not more than what can be extrapolated from protein phylogeny. Moreover, with wrong interpretaion of collinearity between E. ulmoides chromosomes is clear the same misunderstanding have been applied to this section.
3.7 Interpretation of cis-element prediction without experimental verification is just too speculative. This tools are very useful for getting clues for further research. For instance, focusing growing conditions or treatment based in prediction to verify the possible connection. However, using that data to support some possible connections has no meaning.
3.8 That GA3 treatment induce GRF expression it is ok, but with authors experiments relation of GRF expression with leaf growth is not demonstrated. GA3 treatment stimulated growth of organs and clearly internodes, but it does not mean is though GRF, increment of GRF just correlates with GA3 treatment, as it is expected but surely by degradation od DELLA proteins, estimulation of PIFs activities, etc. Probably authors can not transform E. ulmoides or have mutants, not yet at least, but there are other thing can be done. For instance, one of the known effects over growth related with GRF is mainly related with stimulation of cell division, authors can do histological preparations of leaves from non treated plants and treated plants and compare meristems to evaluate differencies in the number of meristematic cells. Although in some conditions GA3 treatment have been also related to meristem cells division, the effect is weaker that the effect on cell elongation of already differenciated cells. If a clear effect is detected, and even not being absolutely and directly connected, it would give a strongest support.
Also, in the case of miRs, evidence is only sequence prediction, and althoug I think is a good prediction, the in vivo verification it is needed to confirm it or all the discussion is just speculation. Once the theorical sequence of the 2 miR candidates are predicted, it would only be necessary to try to find them by using the standar protocols to get small RNA molecules and sequence them. Find them in sequencing data would be a perfect confirmation experiment (even with the possibility of evaluate expression in different tissues to compare with GRF expression) and it is a routinely protocol used in laboratories working with miRs.
Comments on the Quality of English LanguageThere are some mistakes must be revised.
Author Response
Dear Reviewer
We really appreciate you for reviewing the manuscript and giving back your valuable comments which make this manuscript to be more perfect. All of these comments have contributed a lot to improve the quality of this manuscript.
For clarity, all comments and suggestions have been replied one by one and addressed in the revision. The detailed point-by-point responses are listed in the attached file. For clarity, all comments are given with black letters and responses are given with the blue letters below. If there are any other modifications we could make, please feel free to let us know.
Best regards,
Ruoruo Wang

Reviewer 3 Report
Comments and Suggestions for Authors
The authors identified the transcription factor, GRF, which is important for leaf growth in Eucommia ulmoides. Eight EuGRFs were identified and their expression patterns were tested. In addition, treatment with GA3, which induces many GRFs, promoted the growth of E. ulmoides internodes. The data and discussion presented are appropriate and there are no major comments. Please make corrections based on the following minor comments:
Title
grouth --> growth
Methods 2.1.
Please add information on the "soil" used to grow the plant.
Methods 2.1.
Please describe the spraying process in more detail. Did you spray the soil, too? How about the back side of the leaves and stems?
Methods 2.8.
Have the authors checked the specificity of the primers? Please describe that the authors have definitely checked their specificity. In particular, EuGRF1, 2, and 3 should be very similar to each other; therefore, the authors need to carefully check them.
Fig1A
The "5' and 3'" are for the bases, so change them to be used for the amino acids.
Fig6
Please include the number of samples in the legend.
Fig7A
Please indicate the number of growing days and days after treatment in the legend.
If possible, show a photograph of the plant before treatment (day 0) in parallel.
Fig8
Are "Tree scale" and "Time (MYA)" unnecessarily duplicated? Please check.
Specify the meaning of the triangles or remove them.
Author Response

(The authors gave the same response as above.)

Round 2
Reviewer 1 Report
Comments and Suggestions for Authors
Authors have substantially improved the article throughout. They provided responses and followed comments, corrections and suggestions in reviewing article carefully. It is improved than earlier version. However, further improvement in English is necessary. This article can be accepted to publish in Plants
Author Response
Dear Reviewer,
We sincerely appreciate your review of the manuscript once again and for providing us with your comments. Taking into consideration of your suggestion, we have carefully polished the English of our manuscript. Please find the revised manuscript with track changes below. If there are any other changes we could make, please don't hesitate to inform us.
Best regards,
Ruoruo Wang

Reviewer 3 Report
Comments and Suggestions for Authors
All the points raised in the first version of the manuscript have been adequately addressed by the authors. There is nothing more to add to point out.
Author Response
Dear Reviewer,
We sincerely appreciate your review of our response and the revised manuscript once again.
Best regards,
Ruoruo Wang